# Divergent Functions of Rap1A and Rap1B in Endothelial Biology and Disease

**DOI:** 10.3390/ijms26115372

**Published:** 2025-06-04

**Authors:** Ramoji Kosuru, Magdalena Chrzanowska

**Affiliations:** 1Versiti Blood Research Institute, P.O. Box 2178, Milwaukee, WI 53201-2178, USA; rkosuru@versiti.org; 2Department of Pharmacology and Toxicology, Medical College of Wisconsin, 8701 Watertown Plank Road, Milwaukee, WI 53226, USA; 3Cardiovascular Research Center, Medical College of Wisconsin, 8701 Watertown Plank Road, Milwaukee, WI 53226, USA

**Keywords:** angiogenesis, calcium signaling, nitric oxide, inflammation, Rap1A, Rap1B, Rap GEF (Rap1 Guanine nucleotide Exchange Factor), RasGRP3, vascular permeability, vascular immunosuppression, VEGFR2

## Abstract

Rap1A and Rap1B are closely related small GTPases that regulate endothelial adhesion, vascular integrity, and signaling pathways via effector domain interactions, with downstream effectors controlling integrins and cadherins. Although both isoforms are essential for vascular development, recent studies using endothelial-specific knockout models have uncovered distinct, non-redundant functions. Rap1B is a key regulator of VEGFR2 signaling, promoting angiogenesis, nitric oxide production, and immune evasion in tumors while restraining proinflammatory signaling in atherosclerosis. In contrast, Rap1A unexpectedly functions as a modulator of endothelial calcium homeostasis by restricting Orai1-mediated store-operated calcium entry, thereby limiting inflammatory responses and vascular permeability. New insights into Rap1 regulation highlight the roles of context-specific guanine nucleotide exchange factors, such as RasGRP3, and non-degradative ubiquitination in effector selection. Emerging data suggest that isoform-specific interactions between the Rap1 hypervariable regions and plasma membrane lipids govern their localization to distinct nanodomains, potentially influencing downstream signaling specificity. Together, these findings redefine the roles of Rap1A and Rap1B in endothelial biology and highlight their relevance in diseases such as tumor angiogenesis, atherosclerosis, and inflammatory lung injury. We discuss the therapeutic implications of targeting Rap1 isoforms in vascular pathologies and cancer, emphasizing the need for isoform-specific strategies that preserve endothelial homeostasis.

## 1. Introduction

Rap1A and Rap1B are small GTPases closely related to Ras, sharing a conserved G-domain that enables them to function as molecular switches cycling between GDP- and GTP-bound states [1]. Despite sharing identical effector domains, Rap1A and Rap1B differ in their C-terminal hypervariable regions (HVRs), including the CAAX motif required for post-translational prenylation, which governs membrane localization and potentially isoform-specific functions [2,3].

In endothelial cells, Rap1B is the predominant isoform [4]. Both proteins promote endothelial adhesion and barrier function by activating integrins and cadherins through shared effectors and Rho GTPase regulators [5,6,7]. However, recent studies using isoform-specific endothelial knockouts have uncovered distinct roles: Rap1B acts as a positive regulator of Vascular Endothelial Growth Factor Receptor 2 (VEGFR2) signaling, while Rap1A modulates calcium homeostasis and restricts inflammatory calcium nuclear factor of activated T-cell (NFAT) signaling, a previously unrecognized mechanism critical for vascular integrity [8].

Rap1 activity is controlled by guanine nucleotide exchange factors (GEFs), which facilitate the exchange of GDP to GTP and active conformation of Rap1, and GTPase-activating proteins (GAPs), which inactivate Rap1. Epac, the best characterized Rap1 GEF in the endothelium, is directly activated by cyclic adenosine monophosphate (cAMP) and mediates acute Rap1 activation via specific analogs [9,10]. RasGRP3 (CalDAG-GEFIII), a diacylglycerol-sensitive GEF with dual specificity for Ras and Rap1, is expressed in embryonic and angiogenic endothelium and contributes to vascular morphogenesis and diabetes-associated dysfunction [11,12,13,14].

Emerging data reveal that the divergent C-terminal regions of Rap1A and Rap1B, through their differential interactions with membrane lipids, may contribute to isoform-specific effector selection and subcellular targeting [15]. These mechanistic insights, combined with mouse models and human transcriptomic analyses, highlight the differential physiological roles of Rap1A and Rap1B and their implications for vascular pathologies, including inflammation, tumor angiogenesis, and endothelial dysfunction. In this review, we summarize recent advances in Rap1 signaling in endothelial biology, with a focus on new molecular mechanisms, disease relevance, and therapeutic potential of targeting Rap1 isoforms.

## 2. New Insights into Rap1-Mediated Adhesion

GTP-bound Rap1 strengthens endothelial barriers through interaction with effectors such as Krit [16], Ras-Interacting Protein 1 (RASIP1) [17], Radil [18], and Afadin (AF-6) [18,19]. These effectors promote junctional stability by enhancing adherens and tight junction protein interactions and reorganizing the actin cytoskeleton via Rho GTPase modulation [6,7]. In parallel, Rap1-GTP activates integrins through interaction with talin, a key effector. Below, we discuss recent advances in the mechanisms and physiological significance of these pathways.

### 2.1. Spatial Regulation of Rap1 Effector Targeting Towards Integrins or Cadherins

Talin-mediated integrin activation increases integrin affinity for extracellular matrix (ECM) proteins and is essential for cell adhesion and migration [20]. RIAM, a Rap1 effector, facilitates integrin activation by linking talin to integrins [21,22]. In addition, Rap1 binds directly to talin—a step that is critical for integrin activation and particularly for β3 integrin activation [23,24,25,26]. In contrast, SH3 and multiple ankyrin repeat domain protein (SHANK) scaffold proteins (SHANK1, SHANK2, SHANK3) negatively regulate integrin activation by sequestering Rap1-GTP and R-Ras through their Shank/ProSAP N-Terminal (SPN) domains, thereby preventing its interaction with downstream effectors [27]. In endothelial cells, talin and SHANKs compete for Rap1-GTP. Recent studies show that membrane-targeted talin can displace Rap1-GTP from SHANK3, promoting β3 integrin activation [28]. This highlights talin’s dual role: as a direct effector of Rap1-GTP and as an upstream modulator freeing Rap1 from inhibitory SHANK interactions.

Receptor inputs further modulate this balance. Latrophilin-2 (LPHN2), a non-canonical seven-transmembrane G-protein-coupled receptor (GPCR) expressed in endothelial cells [29], dynamically regulates Rap1 signaling [30]. Soluble ectodomains of fibronectin leucine-rich transmembrane protein 2 (FLRT2) bind LPHN2, activating the cAMP–Epac–Rap1 pathway and redirecting Rap1-GTP from talin to SHANK2. This suppresses ECM adhesion and enhances tight junction formation. Silencing LPHN2 diminishes Rap1–SHANK2 interactions, suggesting a receptor-driven reprogramming of Rap1 effectors to favor cell–cell junctions over integrin-mediated adhesion, reinforcing barrier function and limiting Yes-associated protein (YAP)/transcriptional co-activator with PDZ-binding motif (TAZ) signaling [30].

Cortactin serves as a signaling scaffold linking Rap1 to Rac1-mediated actin remodeling. It coordinates Rap1 signaling outputs through effectors such as AF-6, RASIP1, and junctional actin networks [31]. Cortactin associates with VE-cadherin and β-catenin and is necessary for spatial coupling of Rap1 activation to junctional tightening via Epac1. Its absence disrupts Rap1 outputs despite cAMP elevation, suggesting Epac1 requires junctional scaffolding for spatial access to Rap1. Although the GEF responsible remains unidentified, cortactin’s SH3 domain may mediate interactions with Rho GTPase-activating proteins (GAPs) such as Rho family GAP, BPGAP1 [31,32].

### 2.2. Physiological Relevance of Rap1-Integrin Signaling in Postnatal Lung Development

Both Rap1 isoforms enhance VE-cadherin-mediated adhesion. Their combined deletion during development leads to vascular leakiness [33], but postnatally, endothelial-specific deletion does not increase permeability in most vascular beds [34]. However, endothelial Rap1 regulates integrin β1 to support postnatal lung morphogenesis [35]. In endothelial cell-specific Rap1a/b knockout mice (Rap1iECKO), alveolar septation is impaired despite normal capillary development and barrier integrity. This impairment arises from reduced collagen IV—independent of laminin—in the alveolar basement membranes due to deficient integrin β1 activation in Rap1-deficient endothelial cells. Disorganized basement membranes reduce YAP nuclear localization and contractile myofibroblast function, which is essential for postnatal alveolar morphogenesis [36]. Similar phenotypes in endothelial cell-specific integrin β1 knockout mice confirm Rap1’s upstream role in integrin activation. These results parallel findings in *C. elegans*, where RAP-3 activates PAT-2/PAT-3 integrins to promote collagen IV, but not laminin, incorporation [37]. Thus, endothelial Rap1 is critical for ECM assembly required for myofibroblast-driven alveolar maturation [35].

## 3. Isoform-Specific Functions of Rap1A and Rap1B

Studies using endothelial cell-specific knockout mice have revealed distinct roles for Rap1A and Rap1B despite their identical effector domains. Rap1B, the more abundant isoform in endothelial cells, acts as an upstream activator of VEGFR2, regulating angiogenesis, nitric oxide (NO) release, and vascular immunosuppression. In contrast, Rap1A, though less prevalent, plays a specific and essential role in controlling store-operated calcium entry (SOCE), providing a mechanistic explanation for the lung permeability phenotype observed in its absence.

### 3.1. Rap1B as a VEGFR2 Co-Activator in Angiogenesis and NO Signaling

#### 3.1.1. Angiogenesis and VEGFR2 Signaling

Although overall vessel stability is maintained without Rap1 [34,35], Rap1B is specifically required for developmental angiogenesis [38]. Zebrafish models show genetic synergy between Rap1 and VEGFR2, with Rap1B enabling full VEGFR2 activation via integrin αvβ3 [39]. Thus, Rap1B also mediates other VEGF responses, including vascular endothelial growth factor (VEGF)-induced permeability; its endothelial deletion (constitutive Tie2-Cre knockout) protects against VEGF-driven retinal hyperpermeability in early diabetes [4].

In a pathological tumor microenvironment, Rap1B promotes both angiogenesis and immune evasion [40]. Rap1B deletion (inducible VE-cadherin-Cre knockout) impairs VEGF-A-VEGFR2–induced vascular immunosuppression and increases expression of Intercellular Adhesion Molecule 1 (ICAM-1) and vascular cell adhesion molecule 1 (VCAM-1) in response to Tumor Necrosis Factor (TNF)-α, enhancing leukocyte recruitment and CD8⁺ T-cell infiltration and activation, reducing tumor burden in a melanoma model [40] (Figure 1). Together, these findings establish Rap1B as a key modulator of VEGF-driven angiogenesis and immune evasion via VEGFR2 signaling.

#### 3.1.2. NO Release

Beyond angiogenesis, Rap1B is also an essential positive regulator of endothelial nitric oxide synthase (eNOS) activity and NO production, key regulators of vascular homeostasis [41]. Rap1B promotes endothelial junctional complex of platelet and endothelial cell adhesion molecule 1 (PECAM-1)–VE-Cadherin–VEGFR2 complex formation and shear stress-induced VEGFR2 transactivation, which is critical for eNOS activation [34]. Rap1B deficiency (inducible VE-cadherin-Cre knockout) also reduces acetylcholine-induced NO production and vasodilation [42]. Combined deletion of Rap1A and Rap1B exacerbates this defect, leading to hypertension [34].

Mechanistically, Rap1B deletion impairs eNOS Ser1177 phosphorylation, which is essential for NO production [34,43]. Interestingly, Rap1A deletion increases both activating Ser1177 and inhibitory Thr495 phosphorylation, events dependent on calcium signaling [44]. Unlike endothelial cell Rap1B knockout, endothelial cell Rap1A knockout (inducible VE-cadherin-Cre knockout) does not significantly impair vasodilation [43]. This differential regulation suggests that Rap1A restricts calcium-dependent signaling to eNOS, revealing a previously unrecognized role in calcium homeostasis.

### 3.2. Rap1A as a Calcium Homeostasis Regulator via SOCE Suppression

Although both Rap1A and Rap1B promote VE-cadherin adhesion in vitro, postnatal deletion reveals functional divergence. While Rap1B deletion causes no vascular leakage [4], Rap1A deficiency leads to increased lung permeability and inflammation [4,8]. Initial observations showing differential regulation of eNOS phosphorylation by Rap1 isoforms suggested a potential role for Rap1A in calcium signaling [43].

Calcium homeostasis in endothelial cells, critical for endothelial functions, including angiogenesis, permeability, and NO production, primarily depends on SOCE, mediated by calcium release-activated calcium (CRAC) channels [45,46,47]. Agonist-induced 1,4,5-triphosphate (IP_3_) triggers Ca^2+^ release from intracellular stores and subsequent endoplasmic reticulum (ER) Ca^2+^ depletion [48] is sensed by stromal interaction molecule 1 (STIM1), which activates Orai1 channels at ER–plasma membrane junctions [49,50,51,52,53], triggering SOCE. In addition to Ca^2+^ influx, SOCE regulates gene expression via activation of NFAT transcription factors [53,54].

Measurements of intracellular calcium in Rap1A- and Rap1B-deficient cells revealed that Rap1A, but not Rap1B, specifically suppresses the SOCE component of calcium signaling. This regulation occurs at the transcriptional level, as Rap1A deletion increases Orai1 expression, enhancing calcium influx. Normalizing Orai1 expression with partial siRNA knockdown restores SOCE to baseline [8]. In Rap1A-deficient endothelial cells, thrombin induced elevated NFAT activation, an effect reversed by calcineurin inhibition. Lipid nanoparticle (LNP)-mediated delivery of siOrai1 normalizes Orai1 expression and reduces NFAT activity and inflammation in inducible VE-cadherin-Cre Rap1A-knockout (Rap1A^i∆EC^) mice [8] (Figure 2). These findings demonstrate that Rap1A serves as a key negative regulator of endothelial SOCE and inflammation, a novel function in endothelial calcium homeostasis [55].

### 3.3. Distinct Anti-Inflammatory Mechanisms of Rap1 Isoforms

#### 3.3.1. Rap1B, NF-κB, and Vascular Immunosuppression

Although both Rap1 isoforms suppress endothelial inflammation, they do so via distinct molecular mechanisms. Rap1B limits TNF-α-induced nuclear factor (NF)-κB signaling. Endothelial-specific Rap1B deletion (inducible VE-cadherin-Cre knockout) increases NF-κB activation, ICAM-1/VCAM-1 expression, and CXCL11 production, promoting leukocyte adhesion and inflammation [40,56]. As described above (Section 3.1.1), endothelial deletion of Rap1B leads to inflammatory derepression, enhancing CD45⁺ and CD8⁺ T-cell infiltration and activation in tumors, contributing to immune-mediated tumor regression—an effect reversed by CD8⁺ T-cell depletion [40]. Beyond cancer, Rap1B plays a protective role in cardiovascular disease. In atherosclerosis-prone regions, Rap1B-deficient mice (inducible VE-cadherin-Cre knockout) exhibit increased plaque burden and leukocyte accumulation, highlighting its role in maintaining vascular homeostasis [56]. Mechanistically, Rap1B promotes NO production in response to laminar shear stress and suppresses proinflammatory signaling under disturbed flow, positioning it as a key modulator of endothelial inflammation in cardiovascular disease.

#### 3.3.2. Rap1A, SOCE, and NFAT Transcription

Rap1A also plays a central role in regulating inflammation through its control of calcium signaling. As detailed above (Section 3.2), Rap1 suppresses Orai1-mediated SOCE and downstream NFAT signaling. Endothelial-specific Rap1A deletion enhances NFAT1 nuclear translocation and elevates transcription of proinflammatory cytokines, including CXCL1, CXCL11, CCL5, and IL-6. Normalizing Orai1 expression using LNP-mediated siRNA delivery effectively reduces inflammation and vascular leak in vivo [8]. These findings underscore Rap1A’s function as a calcium rheostat that suppresses NFAT-mediated inflammation and its downstream vascular consequences.

## 4. Molecular Basis for Isoform Divergence

### HVR–Directed Membrane Microdomain Targeting

Although Rap1A and Rap1B share an identical effector domain and a conserved Ras-like G-domain (~residues 1–166) [57,58], their functions diverge significantly for reasons that remain incompletely understood. One likely basis for this divergence lies in their C-terminal HVRs, which differ by nine amino acids. These sequence differences influence isoform-specific membrane-targeting properties, contributing to distinct subcellular localization and, potentially, distinct signaling outputs.

Like Ras, Rap1A and Rap1B terminate in a CAAX motif, a site of isoprenylation essential for membrane localization. Unlike Ras, which is farnesylated, Rap1 isoforms are geranylgeranylated at the cysteine residue within their CAAX motifs (Rap1A: CLLL; Rap1B: CQLL) [59,60]. This lipid modification is followed by proteolytic cleavage and carboxymethylation—processing steps analogous to Ras—and is required for plasma membrane association [58,61,62,63].

In Ras biology, lipid modifications and specific interactions between amino acids in the HVR and plasma membrane lipids cooperatively direct Ras proteins to nanoclusters—specialized membrane microdomains essential for efficient and spatially restricted downstream signaling [61,64,65,66,67,68,69]. Given their structural similarity, Rap1 isoforms are hypothesized to follow similar organizational principles.

Recent studies have begun to delineate the lipid-binding properties of Rap1 isoforms, revealing marked differences. Rap1A, like KRAS4B, contains a canonical polybasic domain (PBD) composed of clustered lysine and arginine residues that favor binding to anionic lipids such as phosphatidylserine (PS) and phosphatidylinositol (3,4,5)-trisphosphate (PIP3) and supports nanocluster formation at the inner leaflet of the plasma membrane [15]. In contrast, while Rap1B contains basic residues in HVR, it lacks the canonical PBD, as defined by the high density and clustering of positively charged residues found in Rap1A, and has a greater affinity for cholesterol, phosphatidylinositol 4,5-bisphosphate (PIP2), and phosphatidic acid (PA). These distinct lipid affinities are critical for isoform-specific membrane localization and may direct effector engagement. Emerging models of interleaflet coupling—such as those proposed for KRas4B—suggest that interactions between inner leaflet PS and outer leaflet glycosphingolipids help organize signaling nanodomains [70]. By analogy, Rap1A nanoclusters enriched in PS and PIP3 may participate in similar interleaflet interactions, organizing localized platforms for signaling comprising the Rap1 interactome [15].

This lipid specificity may have functional consequences for downstream targets. For instance, SOCE through the STIM1–Orai1 axis is tightly regulated by lipid microenvironments [71]. STIM1 relies on electrostatic interactions with PI4P, PIP2, and PIP3 at ER–plasma membrane junctions—interactions influenced by the surrounding lipid landscape, including outer leaflet sphingolipids. Rap1A’s selective enrichment in PIP3- and PS-rich microdomains raises the possibility that it may define permissive zones for Orai1 activation, thereby facilitating localized calcium influx in endothelial cells.

Although direct evidence is still needed to determine whether Rap1A organizes lipid nanodomains to control Orai1 activity and whether Rap1B similarly localizes to lipid microdomains that promote VEGFR2 signaling, these emerging lipid-mediated mechanisms offer a compelling framework for understanding the non-redundant roles of Rap1 isoforms in endothelial calcium signaling, inflammation, and barrier regulation.

## 5. Dynamic Regulation of Rap1 Activity

### 5.1. Positive Regulation: GEF Diversity and Functional Contexts

Rap1 activation is mediated by multiple GEFs—including Epac, PDZ-GEF, and C3G—that link distinct upstream cues to context-specific endothelial responses [72,73]. Early studies identified Epac1 as a key cAMP-responsive GEF that enhances endothelial barrier integrity via AF-6, counteracting RhoA-mediated permeability [74,75]. PDZ-GEF has been shown to maintain basal junctional stability, while Epac1 reinforces junctional actin organization under stress [18,76]. C3G, another Rap1 GEF, facilitates barrier recovery following thrombin challenge by inhibiting Rho activity [77,78].

More recently, ArhGEF12 was identified as a non-canonical Rap1A activator in human dermal microvascular endothelial cells [79]. Traditionally recognized as a RhoA-GEF in human umbilical vein endothelial cells (HUVECs), ArhGEF12 selectively activates Rap1A—but not Rap1B or other GTPases—to counteract TNF-induced junctional disruption. ArhGEF12 knockdown reduces basal Rap1-GTP and early Rap1A activation while increasing late-phase RhoA activity, suggesting a temporal GTPase switch [79]. These findings reveal the importance of vascular bed–specific and context-dependent GEF usage in Rap1 signaling.

RasGRP3 (CalDAG-GEFIII) is a member of the RasGRP family GEFs with dual specificity for Ras and Rap1 [11]. Initially identified as a vascular gene responsive to phorbol esters, RasGRP3 is expressed in embryonic and angiogenic endothelium and participates in DAG-sensitive morphogenic signaling pathways [12,13]. It promotes endothelial cell migration and mediates diabetes-induced vascular dysfunction via Ras activation, with its activity driven by elevated DAG levels characteristic of the diabetic environment [12].

Studies in macrophages have shown that RasGRP3 controls their critical functions by acting via Rap1 [14]. However, RasGRP3’s in vivo functions remain incompletely understood, and future studies using endothelial cell-specific RasGRP3-knockout models will be necessary to delineate Ras- versus Rap1-dependent outputs.

### 5.2. Negative Regulation: GAPs, Sequestration, and Inhibitory Scaffolds

Several negative regulators of Rap1 signaling have been identified in endothelial cells. PlexinD1, a neuronal guidance receptor, functions as a Rap1-specific GAP in response to Semaphorin binding [80,81]. Ligand-induced dimerization activates its GAP domain, promoting Rap1-GTP hydrolysis and cytoskeletal retraction—processes essential for vascular pruning and guidance [82]. In Drosophila, PlexinA regulates Rap1 even in the absence of GAP catalytic activity, suggesting additional scaffold-like roles [83].

Independently of Rap1, PlexinD1 also acts as a ligand-independent mechanosensor activated by shear stress [84]. Under flow, PlexinD1 adopts an open conformation that triggers VEGFR2, extracellular signal-regulated kinase (ERK), and eNOS signaling independently of Semaphorin binding or GAP activity [84]. Although Rap1 was not directly examined, Rap1 has been shown to act upstream of VEGFR2 transactivation and endothelial mechanosensory complex assembly [34], suggesting potential crosstalk between PlexinD1 and Rap1 pathways.

TRPM8, the best known as a cold-sensitive ion channel from the melastatin transient receptor potential subfamily, also functions as a Rap1 GTPase inhibitor through a pore-independent mechanism [85]. TRPM8 binds preferentially to inactive Rap1-GDP, retaining it intracellularly and preventing its translocation to the plasma membrane. This sequestration impairs Rap1 activation and downstream β1-integrin signaling, leading to reduced endothelial adhesion, migration, and sprouting. Notably, these effects are independent of TRPM8’s ion channel activity, highlighting a novel non-conductive function of TRPM8 in regulating endothelial behavior via direct modulation of Rap1 [85]. IQGAP1, a scaffolding protein [86,87], promotes pulmonary endothelial cell barrier dysfunction and inflammation in response to lipopolysaccharide by suppressing Rap1/Src signaling. IQGAP1 knockdown restores Rap1-GTP levels and improves barrier integrity [88].

### 5.3. Ubiquitination as a Mechanism for Effector Switching

Emerging work has uncovered non-degradative ubiquitination as a spatially regulated mechanism that modulates Rap1 signaling in endothelial cells [89]. Under laminar shear stress, the E3 ligase WWP2 mediates K31-linked ubiquitination of Rap1A and Rap1B—without altering total protein levels—especially in the descending aorta. This modification occurs within the effector-binding domain of Rap1 and shifts its effector preference from Talin1 (focal adhesions) to AF-6 and RASIP1 (adherens junctions). As a result, ubiquitinated Rap1 enhances junctional stability and suppresses ROS production, promoting endothelial barrier function under flow. Loss of WWP2 or expression of a ubiquitination-deficient Rap1B-K31R mutant mimics Rap1 deficiency, highlighting the physiological relevance of this modification. While both isoforms are ubiquitinated at K31, Rap1B appears to be the dominant isoform mediating WWP2-dependent responses in vivo [89]. This flow-induced effector switch may contribute to anti-inflammatory signaling and vascular protection, especially in the context of atherosclerosis.

### 5.4. microRNA Control of Rap1

MicroRNAs (miRNAs) are small, non-coding RNAs that regulate gene expression by targeting 3′-untranslated regions (3′-UTRs) of mRNAs, leading to degradation or translational repression [90]. While most mechanistic studies of Rap1 regulation by miRNAs have been performed in cancer and hematologic cells, emerging evidence supports their involvement in modulating Rap1 expression in endothelial contexts.

In tumor cells, miRNAs such as miR-149 and miR-101 regulate Rap1 directly or through its negative regulator Rap1GAP. miR-149 suppresses Rap1 expression in neuroblastoma and glioblastoma, where Rap1 promotes proliferation and cytoskeletal remodeling [91,92,93]. In head and neck squamous cell carcinoma, Rap1GAP is repressed via EZH2, itself regulated by miR-101 and miR-26a, linking Rap1 to tumor invasion through a miR-101 → EZH2 → Rap1GAP → Rap1 axis [94,95,96]. Circular RNAs (circRNAs) also contribute to Rap1 regulation by acting as sponges [97,98]: circRNA_103801 and circRNA_104980 modulate Rap1-related signaling pathways in osteosarcoma through sequestration of miRNAs such as miR-370-3p and miR-1298-3p [99,100]. In platelets, Rap1 expression is directly regulated by miR-320c, among others (miR-181a, miR-3621, miR-489, miR-4791, miR-4744), influencing platelet reactivity during storage [101].

In endothelial cells, direct regulation of Rap1 by miRNAs remains less well characterized. In zebrafish and cultured endothelial cells, miR-107 was shown to target both Rap1A and Rap1B, reducing their expression and impairing angiogenic sprouting. This effect was associated with decreased mTORC1– ribosomal protein S6 (Rps6) signaling and confirmed by miRNA pull-down assays [102]. These data suggest a role for miR-107 in controlling Rap1-dependent angiogenic pathways. In a separate study of mechanically stimulated human dental pulp stem cells, several miRNAs (including miR-30e-5p, miR-19a-3p, and miR-154-3p) were identified as modulators of RAP1B expression under mechanical stress. Although performed in non-endothelial cells, the findings suggest that Rap1 isoforms may be subject to miRNA control in the context of mechanotransduction and junction regulation [103].

Current evidence supports post-transcriptional regulation of Rap1 isoforms by miRNAs in multiple cell types. While most studies have focused on tumor or platelet systems, direct targeting of Rap1A and Rap1B by miRNAs has also been demonstrated in endothelial cells. Further work is needed to determine the vascular bed specificity, physiological relevance, and isoform selectivity of these regulatory mechanisms under inflammatory, angiogenic, or mechanical stimuli.

## 6. Therapeutic Implications

### 6.1. Targeting Rap1 in Physiological Disorders: Lung, Retina, and Brain

Epac1-mediated activation of Rap1 was first shown to reinforce endothelial junctional stability and oppose RhoA-driven barrier disruption [74,75,76,104]. Rap1 signaling is now recognized as a central regulator of endothelial barrier function across multiple vascular beds. In pulmonary endothelium, Epac–Rap1 activation enhances junctional integrity and reduces edema in models of lung injury [105,106,107].

However, our recent findings reveal a distinct, non-redundant function for Rap1A in the lung endothelium: the regulation of intracellular calcium homeostasis via the Rap1A–Orai1–NFAT pathway [8]. This mechanism, independent of junctional adhesion, explains the selective increase in lung permeability observed in endothelial cell Rap1A knockout mice [4,8]. Therapeutic targeting of this pathway using siRNA against Orai1 effectively reduced elevated SOCE, mitigated pulmonary edema, and suppressed inflammation. These findings identify Rap1A as a calcium-dependent regulator of endothelial integrity and highlight the potential of modulating Rap1A or Orai1 activity to treat inflammatory lung injury. Further research is needed to refine delivery platforms and enhance specificity for clinical translation.

Rap1 is also an important regulator of vascular integrity in the retina. On the one hand, Rap1 promotes VEGF-induced permeability, and its deficiency prevents junctional disruption in retinal endothelial cells, suggesting a protective role against pathological angiogenesis and leakage in diabetic retinopathy [4]. On the other hand, Rap1 enhances junction formation and barrier function under inflammatory and hyperglycemic conditions [108,109]. Thus, Rap1’s effects are stimulus- and context-dependent, with distinct roles depending on the environment.

Endothelial Rap1 signaling has also been implicated in neurovascular disorders. In a rat model of chronic cerebral hypoperfusion (CCH), the flavonoid compound vitexin restored cognitive function and reduced neuronal damage by activating the Epac1/2–Rap1–ERK pathway [110]. Vitexin treatment reversed CCH-induced downregulation of Epac1, Epac2, Rap1, and phospho-ERK and suppressed inflammation via inhibition of the NLR family pyrin domain containing 3 (NLRP3) inflammasome. While this study supports a neurovascular protective role for Rap1 activation, isoform specificity, and endothelial contributions remain to be clarified.

Complementing this, a network pharmacology and molecular docking study identified Rap1 signaling as a potential therapeutic target of *Astragalus membranaceus* (AM) in vascular cognitive impairment (VCI) [111]. KEGG enrichment analysis highlighted Rap1 alongside phosphoinositide 3-kinase (PI3K)/Akt and MAPK pathways as part of AM’s predicted therapeutic network. Although promising, these findings are computational and require experimental validation to determine the functional relevance of Rap1 signaling—and its isoform specificity—in VCI.

In summary, Rap1 plays a critical physiological role in maintaining vascular integrity across the pulmonary, retinal, and cerebral microvasculature. Its ability to stabilize endothelial junctions, regulate calcium signaling, and suppress inflammation positions it as a versatile therapeutic target for multiple vascular diseases. Future efforts should focus on clarifying isoform-specific mechanisms, developing targeted delivery strategies, and deepening our understanding of Rap1’s physiological roles to fully harness its therapeutic potential.

### 6.2. Pathological Rap1 Signaling in Tumor Vasculature and Beyond

Rap1B has emerged as a context-dependent regulator of tumor vascular function, promoting both angiogenesis and immune evasion through VEGF–VEGFR2 signaling (see Section 3.1.1 and Section 3.3.1). Although it contributes to endothelial NO production and vascular homeostasis [34,42], Rap1B is also essential for VEGF-driven immunosuppressive signaling in tumors [40]. Selective inhibition of Rap1B may sensitize tumor endothelium to inflammatory cues and enhance immune cell infiltration but must be carefully balanced to preserve its physiological roles. Potential strategies include RNA-based inhibition, nanoparticle-mediated delivery, or disruption of Rap1B-specific post-translational modifications or Rap1B interactions with GEFs and effectors [112].

Interestingly, while Rap1B promotes VEGFR2-mediated endothelial anergy and immune evasion, it may also support vascular normalization in VEGF-independent settings. In a murine hepatocellular carcinoma model, lenvatinib enhanced anti-PD-1 therapy by promoting vascular remodeling through a VEGF-compensatory pathway [113]. Specifically, lenvatinib induced the formation of a neuropilin-1–PDGFRβ complex that activated the Crkl–C3G–Rap1 signaling cascade, thereby strengthening endothelial–pericyte interactions. Although Rap1 activation was observed in human endothelial cells in vitro, its contribution to vascular remodeling in the tumor microenvironment remains to be confirmed in vivo. These findings suggest that Rap1 may exert immunosuppressive effects when driven by VEGF but may support vascular normalization and immune activation when engaged by alternative pathways such as PDGFRβ [113].

Supporting its relevance as a therapeutic target, single-cell transcriptomic analyses have identified elevated Rap1B expression in tumor-associated endothelial cells across multiple cancers, including lung, colorectal, and ovarian tumors [40,112,114]. Similar enrichment of the Rap1 pathway has been reported in esophageal squamous cell carcinoma (ESCC) [115]. Rap1 signaling was among the top KEGG-enriched pathways in endothelial cells from lower ESCC, associated with calcium regulation, cytoskeletal remodeling, and cell survival, while upper ESCC endothelial cells showed enrichment in anti-apoptotic and intracellular signaling programs, including increased expression of ARHGEF12—a GEF known to activate Rap1A in specific vascular contexts [115].

Outside of carcinoma, Rap1 signaling was among the top enriched KEGG pathways in infantile hemangiomas based on differentially expressed genes [116], though specific effectors were not detailed. In craniopharyngioma, endothelial cells displayed a distinct pro-angiogenic transcriptomic profile enriched in Rap1, PI3K-Akt, Ras, and Wnt pathways, implicating Rap1 in aberrant neovascularization around tumor epithelial structures [117]. While these associations are derived from in silico analysis of scRNA-seq datasets, functional validation remains essential to confirm the mechanistic and therapeutic relevance of Rap1 signaling in each disease context.

Notably, suppression of angiogenetic Rap1 signaling may be beneficial beyond cancer. In a rat model of collagenase-induced tendinopathy, pathological neovascularization was associated with elevated VEGF and Rap1 expression. Therapeutic suppression of neo-vasculature formation alleviated disease features and was accompanied by reduced VEGF and Rap1 protein levels, along with downregulation of Rap1 pathway genes identified by transcriptomic analysis [118].

In summary, Rap1—particularly Rap1B—emerges as a context-dependent mediator of pathological angiogenesis and immune evasion in cancer and other vascular diseases. While targeting Rap1B offers therapeutic potential, especially in the tumor microenvironment, its essential role in endothelial NO production and vascular homeostasis necessitates caution. A more complete understanding of the signaling context, isoform- and cell-type specificity, and compensatory pathways will be critical for designing selective interventions that disrupt pathological Rap1B activity without compromising its physiological functions.

## 7. Conclusions and Future Directions. Toward Isoform-Specific Therapeutics in Vascular Disease

Rap1 has long been recognized as a regulator of endothelial adhesion through integrin and cadherin modulation. However, emerging evidence reveals that Rap1 signaling extends beyond adhesion to control distinct endothelial functions, including calcium homeostasis, angiogenesis, vascular permeability, and immune regulation. These diverse outputs appear to depend on the selective engagement of downstream effectors, shaped by cellular context, activating receptors, and regulatory mechanisms such as ubiquitination and membrane compartmentalization.

Despite sharing identical effector-binding domains, Rap1A and Rap1B fulfill non-redundant physiological roles. Rap1B acts as a positive regulator of VEGFR2 signaling and endothelial NO production, essential for vascular homeostasis. Its role in promoting tumor vascular immunosuppression makes it a compelling therapeutic target, though interventions must be carefully designed to avoid compromising its protective roles in normal vasculature. In contrast, Rap1A restricts SOCE via the Orai1–NFAT pathway, limiting calcium overload and inflammatory injury in the lung endothelium. This identifies Rap1A as a key regulator of endothelial calcium signaling and barrier function, distinct from canonical junctional pathways.

The molecular basis for this isoform specificity is an active area of investigation. Recent work implicates the C-terminal HVR in guiding Rap1A and Rap1B to distinct plasma membrane microdomains through specific interactions with phosphoinositides and other lipids. This spatial segregation may enable Rap1A to act at ER–plasma membrane junctions, regulating SOCE while positioning Rap1B to support VEGFR2 signaling within raft-enriched domains. Future studies should define how these localization patterns translate into functional specificity in vivo.

Upstream control of Rap1 is also more complex than previously appreciated. While Epac remains a widely used tool to activate Rap1 via cAMP, new GEFs such as RasGRP3 are emerging as physiological regulators. RasGRP3, a dual-specificity GEF for both Rap1 and Ras, contributes to vascular morphogenesis and diabetes-associated endothelial dysfunction. This signaling integration and disease relevance warrant further mechanistic and functional exploration.

In parallel, bioinformatic analyses have implicated Rap1 signaling in the endothelial tip/stalk cell specification [119] and in diseases such as vascular cognitive impairment and tendinopathy. While these in silico findings suggest new biological and pathological roles for Rap1 function, they require direct experimental validation.

Therapeutically, Rap1 isoforms present distinct opportunities for intervention. Targeting Rap1B in the tumor microenvironment may enhance immune cell infiltration and disrupt vascular immunosuppression while preserving normal endothelial function by avoiding systemic inhibition. Conversely, modulating Rap1A–Orai1 signaling offers a promising strategy to mitigate inflammatory lung injury and restore barrier integrity. Rap1 pathway components may also be leveraged in conditions such as diabetic retinopathy, cerebral hypoperfusion, and tissue remodeling, provided isoform- and context-specific effects are clearly understood.

Altogether, Rap1 signaling integrates receptor cues, membrane microdomain localization, and isoform-specific effector selection to coordinate endothelial behavior across physiological and pathological contexts. A deeper mechanistic understanding of these processes will be essential to develop selective therapeutics that correct disease-associated dysfunction while preserving vascular homeostasis. Whether these mechanisms are conserved across all vascular beds or exhibit tissue-specific variation remains to be determined. Future work should define how Rap1A and Rap1B are differentially regulated in distinct endothelial subtypes, particularly across organ-specific microvascular environments.

## Figures and Tables

**Figure 1 ijms-26-05372-f001:**
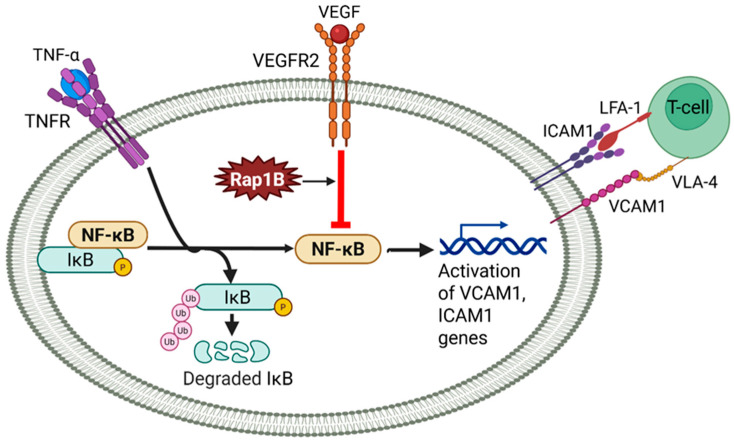
Endothelial Rap1B is essential for VEGF-induced vascular immunosuppression. Endothelial Rap1B suppresses NF-κB–driven expression of cell adhesion molecules (VCAM1, ICAM1), limiting T-cell adhesion and recruitment. In the tumor microenvironment, VEGF–VEGFR2 signaling enhances this immunosuppressive effect through Rap1B, promoting endothelial quiescence and barrier to immune infiltration.

**Figure 2 ijms-26-05372-f002:**
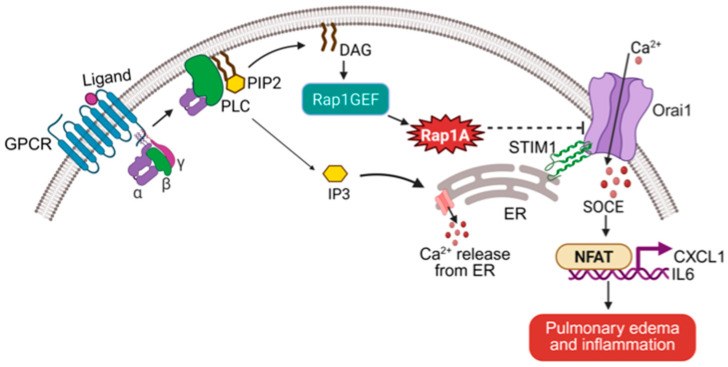
Rap1A restricts Orai1-mediated calcium entry to suppress inflammatory signaling in endothelial cells. The schematic illustrates the Rap1A–calcium signaling axis in endothelial cells. Upon GPCR activation, phospholipase C (PLC) hydrolyzes PIP2 into diacylglycerol (DAG) and inositol 1,4,5-trisphosphate (IP3), a diacylglycerol-sensitive guanine nucleotide exchange factor that activates Rap1A. Activated Rap1A suppresses store-operated calcium entry (SOCE) by limiting Orai1 activation and expression. In the absence of Rap1A, increased calcium influx via Orai1 promotes nuclear translocation of NFAT and transcription of proinflammatory genes such as CXCL1 and IL6, leading to pulmonary edema and inflammation. This pathway identifies Rap1A as a critical negative regulator of calcium-driven endothelial inflammation.

## Data Availability

No new data were created or analyzed in this study. Data sharing is not applicable to this article.

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
