# Peer review of "Divergent Functions of Rap1A and Rap1B in Endothelial Biology and Disease"

_ijms, 2025, doi:10.3390/ijms26115372_

Round 1
Reviewer 1 Report
Comments and Suggestions for Authors
Dr. Chrzanowska is a well-recognized investigator in the fields of RAP1 and vascular biology and is highly qualified to summarize the development of RAP1 in endothelial biology and disease. The review is well written and well structured. It is particularly commendable that the authors highlight the potential role of the C-terminal membrane anchors of RAP1A and RAP1B in determining their isoform-specific functions. I strongly recommend acceptance of this review following minor revisions:
-
Lines 48–51: Please define GAPs and GEFs and briefly describe how they regulate small GTPases in general.
-
Section 3.3.2 appears somewhat redundant, as the regulation of SOCE by RAP1A is already discussed in section 3.2.
-
Line 253: Although RAP1B has a lower net charge, its C-terminus also contains a PBD with three lysines and one arginine.
-
Line 258: "KRas4G" should be corrected to "KRAS4B."
Author Response
Comment 1. Lines 48–51: Please define GAPs and GEFs and briefly describe how they regulate small GTPases in general.
Response 1: This change was made.
Comment 2. Section 3.3.2 appears somewhat redundant, as the regulation of SOCE by RAP1A is already discussed in section 3.2.
Response 2: To reduce redundancy, we removed overlapping mechanistic detail from Section 3.2 while retaining it in Section 3.3.2, which focuses specifically on the distinct inflammatory roles of Rap1A and Rap1B.
Comment 3. Line 253: Although RAP1B has a lower net charge, its C-terminus also contains a PBD with three lysines and one arginine.
Response 3: We thank the reviewer for the clarification. We have revised the text to distinguish between the presence of basic residues and a canonical polybasic domain. Specifically, we note that while Rap1B contains basic residues in its hypervariable region, it lacks the canonical PBD found in Rap1A—defined by a high density and clustering of positively charged residues—and exhibits distinct lipid-binding preferences.
Comment 4. Line 258: "KRas4G" should be corrected to "KRAS4B."
Response 4: We corrected the text accordingly.
Reviewer 2 Report
Comments and Suggestions for Authors
The review by Kosuru and Chrzanowska focuses on recent advances in Rap1 siganling, specifically differences in signaling between Rap1a and Rap1b isoforms that have recently come to light. Overall, the review does a good job of introducing the topic and highlighting the known differential effects of the two isoforms. The conclusions are reasonable, and the review summarizes the field in a way that highlights the remaining questions. The manuscript is well written and easy to read.
The manuscript could be improved by:
- Including a discussion of regulation of Rap by microRNA and how that fits into the pathways described.
- Including a discussion of which mechanisms may be tissue or endothelial cell type specific- or whether that is known.
Author Response
"The manuscript could be improved by:"
Comment 1. Including a discussion of regulation of Rap by microRNA and how that fits into the pathways described.
Response 1: We thank the reviewer for this suggestion. In response, we added Section 5.4 summarizing miRNA regulation of Rap1 in tumor, platelet, and endothelial contexts, and how it intersects with pathways discussed in the review. We also highlight areas where endothelial- and vascular bed–specific regulation remains to be determined.
Comment 2. Including a discussion of which mechanisms may be tissue or endothelial cell type specific- or whether that is known.
Response 2: We appreciate this thoughtful suggestion. While our review is focused on Rap1 isoform-specific signaling in endothelial cells, we agree that the extent to which these mechanisms differ across vascular beds remains an important area for future research. We have added a clarifying statement in the concluding section (Section 7) to acknowledge this knowledge gap and to emphasize the need for further studies to determine how Rap1A and Rap1B signaling may vary across organ-specific endothelial subtypes and tissue contexts.